# Three Quarks for Hypersexuality Research

Piet Van Tuijl *, Peter Verboon  and Jacques J. D. M. Van Lankveld

Department of Theory, Methods and Statistics, Faculty of Psychology, Open University of the Netherlands, 6419 AT Heerlen, The Netherlands
* Correspondence: piet.vantuijl@ou.nl

**Abstract:** In some areas of sex research, we note room for methodological improvement of research practices. In particular, in the field of hypersexuality research, where societal norms have been influential, methodological improvement might advance goals of objectivity in research. We propose that hypersexuality research should: firstly, take into account relevant subpopulations; secondly, use Item Response Theory (IRT) to construct item banks for measurement instruments; and, thirdly, measure sexual desire and related important constructs where and how they play out—in daily life, changing from moment to moment. We show that comparing relevant subpopulations can lead to depathologizing normative, but highly frequent, sexual behavior. Using IRT can lead to more precise measurement instruments by assessment of characteristics of individual items. Measuring sexual desire as an inherently fluctuating process in everyday life, and as part of emotion regulation processes, can direct research towards relevant associations other research methods might miss. Bringing into practice our three proposals for improvement can procure a number of advantages. We illustrate these advantages mainly for the field of hypersexuality research, but our suggestions might also be beneficial for sex research in general.

**Keywords:** hypersexuality; item response theory; experience sampling methodology; validity



## 1. Introduction

Sex research has been closely, but not unequivocally, related to societal views and perspectives on sexuality. While it is one of the goals of scientific research to be as objective as possible, objectivity in sex research might be hard to attain. In particular, in hypersexuality research, opinions have gotten mixed with research designs and outcomes [1] Methodological rigor can be used to counter the influence of biased opinions. In the current article, we present three proposals to increase methodological rigor in hypersexuality research. We hope that with our proposals, problematic hypersexuality (PH) can be studied in more depth. We define PH as distress and negative consequences due to hypersexual urges and behavior to the extent that seeking help is considered by the individual [2,3]. (For an in-depth definition of PH, see Figure 1) Furthermore, by applying our suggestions, conditions that might seem to resemble PH at first can be distinguished from it and, thus, depathologized. We specifically think of Non-problematic Hypersexuality (NH) here, which we define as the experience of hypersexuality without major distress. We hope that our suggestions will render hypersexuality research more independent of opinion and will reduce the influence of societal norms on hypersexuality research.

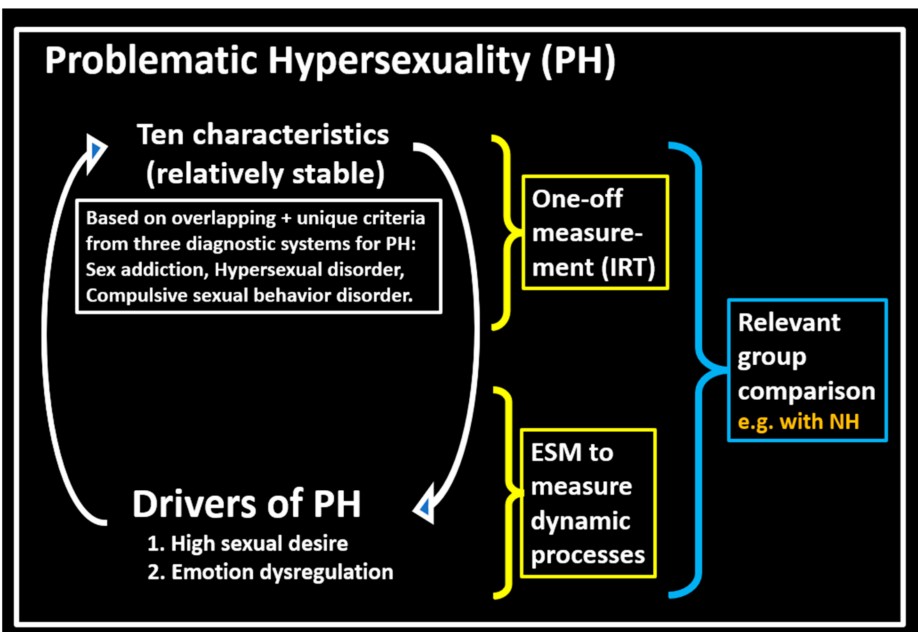

**Figure 1.** Decentralized Construct Taxonomy (DCT [4]) of Problematic Hypersexuality. Model of the DCT definition for PH (https://psycore.one/construct/?ucid=problematicHypersexuality_7mm4hr4f, accessed on 31 January 2023). This DCT implies an extensive, explicit, testable and preliminary definition for PH and includes remarks on how to measure PH. Ten relatively permanent characteristics of PH are seen as driven by high sexual desire and emotion dysregulation in this definition. Stable traits are best measured with IRT-validated one-off instruments, while dynamic processes (the drivers) are best assessed with ESM techniques. IRT and ESM research should be undertaken with relevant group comparisons in mind.

The improvements we propose for hypersexuality research are threefold. Firstly, we propose that researchers award more attention to the comparisons of relevant subpopulations in hypersexuality research. Comparisons of relevant subpopulations can show differences between groups that are quite similar in some aspects, but different in crucial other ways. Such comparison will provide insight into the uniqueness of certain characteristics of PH. We note that in this article, we often discuss the following two subpopulations as relevant to compare: NH and PH. This is because these groups are quite similar, but have one important distinguishing feature: the level of distress that is experienced due to hypersexuality. Therefore, the other differences between these groups might show best what sets PH apart, from NH and from other conditions that are less similar to PH. However, this is not the only relevant comparison, and others might be important, as well. Besides discussion of the advantages of comparing relevant subpopulations, we offer some examples of studies where this has received little attention, showing how not including comparisons of relevant subpopulations might have resulted in misleading interpretations. Secondly, we propose to use Item Response Theory (IRT) analyses for the purpose of constructing and validating measurement instruments for PH, and we will show how this will bring psychometric advantages over currently common validation practices in hypersexuality research (i.e., factor analysis). As IRT allows for the assessment of the discriminative value of separate items, evaluation of the importance of separate criteria for PH becomes possible. In addition to discussing the advantages of IRT, we provide a short tutorial on IRT in the Supplemental Material File S1. Thirdly, we propose to measure sexual desire and behavior where it plays out: in everyday life, fluctuating from moment to moment. As ecological and construct validity might best be served when sexual desire is measured as "a dynamic or a changeable phenomenon" ([5], p. 280), we propose the experience sampling method [6,7] as an additional method to collect data on hypersexuality. We note that hypersexual desire and behavior can be investigated with ESM techniques as part of

a spectrum of emotion regulation [8], which allows for the study of reciprocal effects of mood and hypersexuality, potentially uncovering a vicious circle underlying PH. The three improvements we propose, the three quarks for hypersexuality research, we offer together with some considerations on their limitations. To conclude this introduction, we want to mention that our three proposals are methodological in nature, but can in combination, we believe, add substantive perspective to hypersexuality research.

## 2. Compare Relevant Subpopulations in Hypersexuality Research

A call has been made for investigating relevant subpopulations in hypersexuality research [9] which we underwrite, but we want to add that an essential part of such research should target demarcations between such subpopulations. Though descriptions of subpopulations in themselves can be useful, it is mostly by comparative investigation that the most cue-valid [10] or unique characteristics of these subpopulations can be uncovered. Not considering relevant subpopulations can lead to biased descriptions of populations, as we will show. Furthermore, when relevant subpopulations are not discerned in scale construction, this can lead to biased instruments to measure PH. Currently, some measurement instruments are used for assessing PH that were constructed by comparing a clinical PH population with the general population [11], while other instruments were constructed without any subpopulation comparison at all [12]. We will show how such practices can lead to misleading conclusions, as they might result in cutoff values that seem to be too high, thus suggesting a high percentage of false positives [13]. This can be potentially damaging, in particular for NH individuals, as they might be most at risk of being pathologized by unwarranted conclusions regarding their sexual conduct. Therefore, we want to raise awareness of the advantages of comparing relevant subpopulations in hypersexuality research. There are three advantages we here present: (1) Relevant group comparisons can avoid unwarranted conclusions based on seemingly statistically clear results; (2) Relevant group comparisons can aid in uncovering characteristics of PH; and (3) Awareness of relevant group comparisons invites researchers to consider aspects that might be missing from their current study, and thus inspire research into fertile new directions.

The first advantage of comparing relevant subpopulations is that unwarranted conclusions can be avoided. In PH research within a general population sample, often only a small percentage is affected by it ([14]: 3.5%; [15]: 0.7%; [16]: 1%; and [17]: 1.2%). To reach conclusions based on such small percentages actually afflicted by PH might be misleading if different subpopulations within the hypersexual part of the sample are not taken into account. Especially when NH and PH are not teased apart, such conclusions can tend to corroborate common norms and general opinions about hypersexual behavior. There is a risk of pathologizing frequent sexual behavior as problematic, while individuals do not experience it as such. In a study using a largescale sample that assessed correlates of hypersexuality in the general population, it was shown that higher levels of impersonal sex or masturbation were associated with more negative health indicators [18]. Although this conclusion was statistically correct, we illustrate in Figure 2a–d how it might still have been unwarranted. Consider, as an example, a general population sample in which sexual desire and distress are measured. It consists of a majority of people without high sexual desire and a minority of people that do experience high sexual desire. This minority consists partly of people experiencing increased levels of distress due to hypersexuality, and partly of "healthy controls who report frequent sexual activity in non-problematic ways" ([19], p. 46). Consequently, the high end of the sexual desire continuum will show increased levels of distress due to the PH part of the hypersexual part of the sample (Figure 2a,b), and the general analysis will lead us into thinking that higher sexual desire is positively associated with distress. Yet, if one limits the study to only those showing high sexual desire, as has been done in a paucity of studies [3,14,20], then the positive association between high sexual desire and distress disappears, and the characteristic of high sexual desire does not discriminate between PH and NH anymore (Figure 2b,d). The discriminative value of a characteristic for PH should, therefore, first be established by comparison of a PH

with an NH sample. Furthermore, people with lower levels of sexual desire (the green dots in Figure 2a) will not visit a sexologist or GP for hypersexuality, as they do not doubt themselves in this respect. For people experiencing hypersexuality, however, it can be highly relevant to know if they are "in trouble" and for them is of the utmost importance that their GP or sexologist can make a sound assessment. Therefore, it is firstly within groups of NH and PH individuals (the blue and red dots in Figure 2a,c) that research into discriminating characteristics should be undertaken, as it is within these subpopulations that assessment of PH takes place and its conclusion is most impactful. The restriction of the range of sexual desire to only its higher levels, propagates a method that is usually considered a threat to validity [21]. However, in this case, restriction of range leads to a more relevant comparison of subpopulations than the comparison of the general population to a hypersexual sample. This shows that both for theory, as well as for clinical practice, comparison of relevant subpopulations is consequential and can help to avoid unwarranted conclusions on hypersexuality.

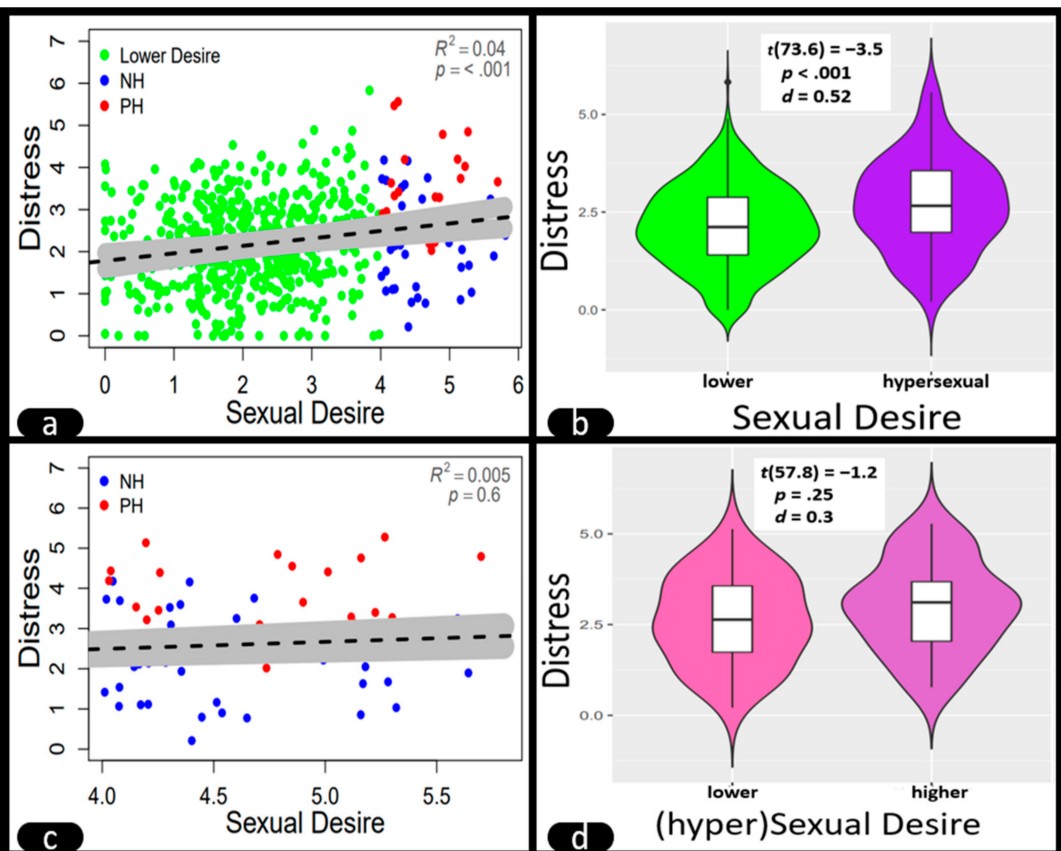

**Figure 2.** Comparing relevant subpopulations. (**a**) Scatterplot of sexual desire and distress in a general population sample. Part of this sample can be considered as PH (=red) or NH (=blue). Sexual desire is positively associated with distress. (**b**) Violin and boxplot comparing lower sexual desire and the hypersexual desire subsamples on levels of distress. The hypersexual sample (PH and NH) experiences significantly more distress than those with lower sexual desire levels. (**c**) Scatterplot of sexual desire and distress in the hypersexual subsample. Half of this sample can be considered as PH and the other half as NH. Sexual desire is not significantly associated with distress. (**d**) Violin and boxplot comparing lower (hyper)sexual desire and higher (hyper)sexual desire samples in the overall hypersexual sample. There is no difference in distress levels between both samples.

The second advantage of comparing relevant subpopulations is that they can aid in un-covering what the unique characteristics of PH are. It is not enough to characterize a group by describing it, although this can undoubtedly provide valuable information (e.g., [22]).

What is needed on top of that are contrasts with relevant other groups. Especially when those other groups are similar to PH in certain ways but differ in important other ways, the contrast will provide perspective on what are unique characteristics of PH and what sets PH apart from other conditions. Finding such contrasting relevant other groups is tantamount in developing measurement instruments. If one considers, for instance, the item "Sex has been the most important thing in my life" (Compulsive Sexual Behavior Disorder—19 scale; [23], item 4), one might suspect this item does not distinguish well between PH and NH groups. That the item might discriminate well between an outpatient clinical PH group and the general population does not suffice to validate its differentiating value in a more relevant situation where the diagnostic differences are less clearcut [3]. In the construction and validation of a measurement instrument for PH, the comparisons of relevant subpopulations are an important step in determining what set of characteristics pertain uniquely to PH. Yet, in validation research, sometimes only general population samples are used [12], and if clinical groups are used, these are either compared to a general population sample [11] or to no other groups at all [19]. This means that in the development of measurement instruments for PH, often relevant comparisons of subpopulations are lacking. Comparisons that have only secondary value, as those comparing a general population with a clinical PH sample, will heighten chances of finding significant differences between the samples. However, these less relevant comparisons might also lead to the construction of scales that do not discriminate well between subpopulations where such discrimination is most relevant, thus increasing the risk of false positives and leading to high estimates of PH in the general population (e.g., 15.8% for men and 5.7% for women; [24], p. 2210, measured with the Hypersexual Behavior Inventory). Using comparisons of relevant subpopulations, we can better determine what uniquely characterizes PH, and this allows researchers to integrate cue-valid determinants of PH in future studies.

The third advantage of comparing relevant subpopulations is that awareness of such comparisons might inspire researchers to consider aspects that are currently missing, but if included in their research, might open new venues for investigation. As an example, we refer to a recent study that showed how oxytocin plasma levels of men suffering from PH significantly decreased after group therapy [25]. The conclusion of the authors was that men afflicted by PH show a hyperactive oxytonergic system to attenuate stress, and that oxytocin might be a potential biomarker for PH. However, in this study, no NH controls were included. Had the authors considered comparing PH and NH subpopulations, it might have inspired thinking about the levels of oxytocin in NH individuals. If for NH the same high levels are found as for PH, this will refute an association between oxytocin and distress in hypersexuality. Therefore, currently no conclusion can be drawn regarding the association between oxytocin and distress due to PH, as it is unknown what these levels are in the NH group. If these levels are much lower in the NH group, high oxytocin levels might more convincingly be considered an indicator of PH. These considerations—inspired by the notion of relevant group comparisons—will add nuance to research into the role of plasma oxytocin levels in PH and NH and might also show that oxytocin has different functions in different populations. In short, considering the comparison of relevant subpopulations can inspire research to look for the necessary nuance that is needed to understand the phenomenon of hypersexuality more deeply.

In conclusion of this first quark for hypersexuality research of comparing relevant subpopulations, we consider the two dimensions suggested for PH by Kingston [2]: (1) high sexual desire; and (2) emotion dysregulation with regard to sex. A common mistake when theorizing about hypersexuality is to think that necessary aspects of PH, such as high sexual desire, are what characterizes PH in its essence. Both of the dimensions of PH we consider to be necessary for PH to occur–and we conceptualize these as drivers of PH (Figure 1). However, none of these dimensions in itself is sufficient for PH to be the case. Neither can they automatically be used as cue-valid indicators, as we have shown in this first quark. Prolonged high sexual desire might perfectly well be experienced as non-problematic, and emotion dysregulation can occur in numerous other conditions. We think it has been a

mistake to conceptualize high sexual desire as a risk for PH [26]. The suggestion that, as with alcoholic beverages, there is a certain level of high sexual frequency beyond which one is certainly at risk (e.g., >6 orgasms per week, [27]) is not warranted, nor supported by largescale research [28]. Equating "excessive sexual drive" with a dysfunction (e.g., [29], code F 52.8) would be cutting corners and would lead to high percentages of false positives for PH, especially when research uses analytical and psychometric techniques that do not include comparisons of relevant subpopulations, as we have shown. Such practices increase the risk of pathologizing unproblematic sexual feelings and behavior. We gave the example of a study that indicated a correlation between high sexual desire and negative outcomes [18], but, as exemplified in Figure 2a, such a conclusion might have come about because relevant subpopulations were not included in the design.

We like to mention three limitations that come into play when one considers the comparisons of relevant subpopulations. First, it might not be clear beforehand what the relevant subpopulations to compare with are. We strongly suggest to consider comparing PH with NH subpopulations (in this we need to consider our own bias), but this is not the only comparison that might be of value in hypersexuality research. To find the comparison that is most relevant in a given study should be a matter of careful consideration. This leads to a second limitation: it is not clear beforehand how far one needs to go in discerning separate subpopulations ("subsubpopulations"). If we consider Figure 2b, which shows the relation between sexual desire and distress for the hypersexual population, we might be able to distinguish within the PH group a subpopulation that compulsively watches porn and a group that is dependent on chemsex. These two subsubpopulations might show very different correlations between sexual desire and distress. There is a large heterogeneity in PH presentations [30,31], and this heterogeneity taps into the "crud factor"—the inexhaustible source of possible other factors that can be related to a certain outcome [32]. However, this should not lead to relativism and does not exempt researchers from searching for the most relevant comparisons. Finally, we mention a third limitation of this first quark: when considering some hypersexual subpopulations, even the broad structure of the two dimensions for PH (high sexual drive and emotion dysregulation) might not apply. This might, for instance, be the case for sexual offenders, where not so much the presence, but rather the absence of distress might correlate positively with hypersexuality [33]. It is beforehand not clear how hypersexuality can be studied in such outlier subpopulations. These three limitations considering the heterogeneity of PH presentations seem to defy the development of a clear research plan for PH. However, despite these limitations, and in conclusion, we think that awareness and search for comparisons of relevant subpopulations can be an important step forward in hypersexuality research and might be of substantial importance to avoid unwarranted conclusions on PH, aid in finding unique characteristics for PH and inspire nuanced research into hypersexuality.

### 3. Use Item Response Theory to Develop Problematic Hypersexuality Measurement Scales

The comparison of relevant subpopulations, discussed in the previous section, needs to be undertaken with well-validated instruments. Scales in hypersexuality research are usually constructed with EFA and CFA (e.g., [11,16,19]). We propose adding IRT techniques to these methods to construct and validate instruments for hypersexuality research. There are three advantages of IRT we would like to present: (1) IRT integrates specific answer patterns of respondents in the construction of a scale, which allows for locating not only respondents on that scale, but also items; (2) IRT allows for the testing of individual items of measurement instruments as separate criteria for PH; and (3) IRT allows for building up an item bank to measure problematic hypersexuality.

The first advantage of IRT is that this technique integrates specific answer patterns of respondents in scale construction, and this provides advantages over standard techniques. In scale construction where only EFA and CFA are used, specific answer patterns are not taken into account, and sum scores are used to assess the individual's scale score [34]. With

sum scores, respondents are ordered based on their scores, but items are not ordered on this scale; every item is seen as equally important. As a consequence, items that generate the response "a lot" very often (for instance, the item "Do you think of sex?" in a hypersexual sample) have the same weight as items that generate a more evenly spread distribution of responses (for instance the item "Has sex become a problem for you?" in a hypersexual sample). The IRT model integrates different answer patterns and, thus, allows for both items and respondents to be located on the same underlying latent trait scale (see Figure 3). The scale score is not a sum score, but a latent trait score with each item individually weighted. Not only does this seem fairer, it also allows for the assessment of important item characteristics, and this brings us to the second advantage of IRT.

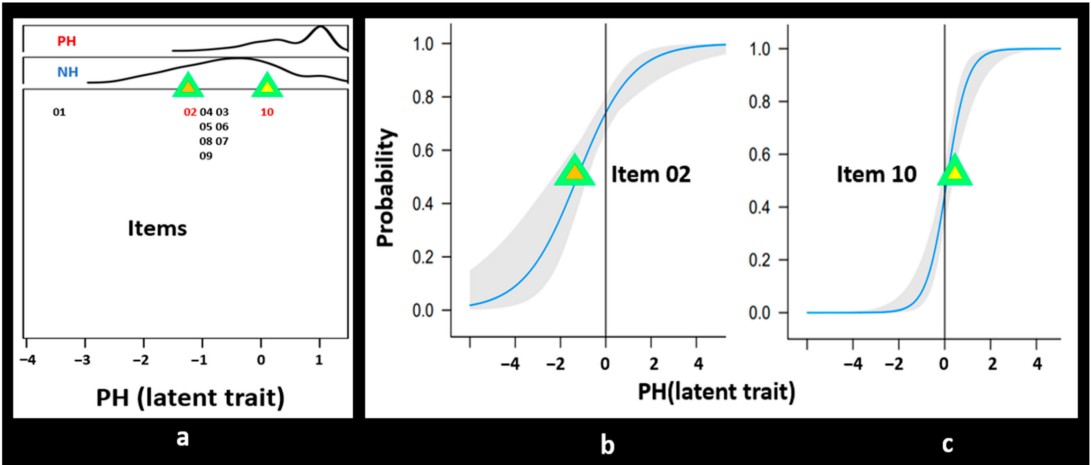

**Figure 3.** IRT output. (**a**) Wrightmap of 10 items that measure PH (based on simulated data). Both participants (divided in PH and NH groups) and items are located on the latent trait of PH. (**b**) Item Characteristic Curve of item 02 "Doing something sexual helps me cope with stress". (**c**) Item Characteristic Curve of item 10 "My sexual appetite has gotten in the way of my relationships". A short tutorial on IRT analyses based on these data can be found in the Supplemental Material File S1.

IRT measures certain characteristics of individual items in such a way that they can be tested as separate criteria for PH. In particular in hypersexuality research, this could give extra impetus to instrument construction and validation because there still is discussion about a number of criteria that might or might not define PH (e.g., withdrawal or tolerance symptoms, [35]). Because IRT orders individual items on the same underlying latent trait scale as participants, it can be estimated if an item can distinguish well between two populations. If we regard the frequency distribution of latent trait scores for the PH and NH samples in Figure 3a, we can deduce that item 02 cannot distinguish well between the two subpopulations due to its location on the latent trait scale. Furthermore, we see in Figure 3b that the item characteristic curve (ICC) for item 02 is not so steep, indicating that this item has less discriminative power than item 10, which has a steeper ICC. Item 02 will often be scored as a "yes" by respondents from both groups. This item might be "Doing something sexual helps me cope with stress" (Hypersexual Behavior Inventory-19, item 13, [19]), as both people from the NH and PH subpopulations can be expected to use sex to cope with stress or other negative mood states [36]. Item 10 might better discriminate between NH and PH than item 02, as it is located in the middle of both frequency distributions, and its curve is also steeper. Item 10 might be "My sexual appetite has gotten in the way of my relationships" (Sexual Compulsivity Scale, item 1, [37]). This item is endorsed in our example by a large part of the PH group and is much less endorsed by the NH respondents, which implies that the NH group suffers much less from negative consequences, e.g., break-up, than the PH group. Item 10 might be used both for discriminative purposes and can also help to evaluate severity [38] and development over time [39] of PH. As our example presented in Figure 3 shows, IRT can be used to gauge cue validity of characteristics of PH.

It can be investigated if certain items—representing these characteristics—are for instance located in the middle of NH and PH subpopulations. If that is the case, these characteristics are what uniquely determine PH and set it apart from NH. Finding a whole set of such characteristics, and the items representing them, allows for the buildup of an item bank, and this brings us to the third advantage of IRT.

The buildup of an item bank implies that a large set of items measuring PH can be brought together in one item bank that can function as a reservoir for Computerized Adaptive Testing (CAT) or shorter questionnaires for diverging subpopulations. All instruments stemming from the item bank are scored on the same underlying latent trait [40]. Currently, a number of separate questionnaires stemming from different diagnostic perspectives represent fixed sets of items to measure PH. The most useful items of previous scales developed for PH could be detected with IRT and could then be integrated in an item bank. Calibration and evaluation plans have been set out for such development of health-related scales (e.g., [41], p. 24 for the Patient-Reported Outcome Measurement Information System (PROMIS, [40]). In how far these methods can be applied in the development of a dynamic, large item bank for hypersexuality remains to be seen. However, the option of using PH instruments in unison has been voiced previously: "At this stage in the research, agreed-upon combinations of scales to tap different dimensions of HD could represent an optimal compromise" ([42], p. 14). To such a merging strategy that encompasses all instruments for PH, IRT not only adds the possibility to develop CATs or short forms for (sub)populations, IRT validation also allows for the possibility to keep on developing the item bank, by either adding new items to the initial set or by investigating new (sub)populations. In this way, the buildup of an item bank for PH becomes "a perpetual work in progress" ([40], p. 9) that consists of a back-and-forth between instrument calibration and data collection. In short, using IRT might inspire a sea-change in validation research for hypersexuality assessment, as it allows for a continuous development and finetuning of measurement instruments and thus engenders continuity in the collection of information on PH.

In conclusion of this second quark for hypersexuality research—the use of IRT to construct and validate measurement instruments—we proffer an ideal picture of the development of an item bank to assess PH. Despite the accepted diagnosis for PH of Compulsive Sexual Behavior Disorder (CSBD [43]), there still remains ambiguity about what characterizes PH. Moving from the diagnosis of Hypersexual Disorder, rejected for inclusion in the DSM-5 [44], to the CSBD diagnosis, aspects of emotion dysregulation were abandoned as criteria ([45], p. 379 criteria A3 and A4; [43], guidelines concerning shame and guilt, p. 109). This implies that emotion dysregulation—one of the two dimensions of PH, according to other authors [2,33]—is interpreted as a less important aspect of PH by the authors of the CSBD diagnosis. Additionally, given there are only four criteria in the CSBD diagnosis, one might ask if this covers all aspects of the phenomenon. Furthermore, there are even more aspects of PH put forward that need consideration, such as those stemming from the sex addiction perspective on PH (e.g., withdrawal and tolerance symptoms [46]). With IRT, the cue-validity of these and other criteria can be assessed in different subpopulations, and this can lead to a more definite set of items for questionnaires and a more extended set of characteristics for PH. There is no need for adamant adherence to theory or etiology if methods to test theory improve enough. What needs to be done then is to test the theory. In particular, if this can be done with IRT, where characteristics of PH can be tested as items of an item bank, sufficient nuance can be attained to develop instruments for evaluative and discriminative purposes [38,39]. The finetuning that IRT allows for can be an important asset for hypersexuality research, beneficial to the aim of reaching conceptual clarity.

We would like to mention three limitations to the use of IRT to construct and validate instruments for PH. First, IRT applies only to a set of items that represent one factorial dimension, or two dimensions at most if the second factor is much less important than the first [41]. Current instruments for PH usually consist of three or more dimensions (e.g., [23]). IRT could still be used to assess item characteristics for the separate factors, but what this would actually signify for the measurement of PH is not clear beforehand.

A second limitation, associated with unidimensionality, is unclarity about what reference group to use. In the PROMIS set-up, disease groups are regarded relative to the general (United States) population. However, we would not propose comparison to a general population in the case of PH item bank development, because in addition to our arguments regarding relevant comparisons, we would expect that the general population to show large floor and ceiling effects, thus biasing the overall investigation. This suggests a third limitation: certain dimensions of PH lie outside the scope of IRT, as these dimensions might not be captured well with one-off scales. The Coping-scale [19] might constitute an example of this. This scale intends to measure aspects of emotion dysregulation with regard to sex. However, research seems to refute that sex used as coping is a cue-valid characteristic of PH (e.g., [24,36]). As one author put it: "Using sex to handle stress and mood disturbances is not much different from using exercise, prayer, or meditation for the same purpose." ([47], p. 228). The conclusion seems warranted that emotion dysregulation is not an issue in PH, which is in line with the guideline accompanying the CSBD diagnosis concerning shame and guilt [43]. However, one needs to consider the dynamic processes involved in PH and investigate those with different methods because these processes (e.g., sexual desire and emotion dysregulation) might not be well traceable with one-off instruments. To conclude, and despite these three limitations of IRT, we promote the use of IRT methods, as these can help determine unique characteristics of PH and, thus, offer methodological improvement to the development and validation of measurement instruments for hypersexuality research.

### 4. Assess Hypersexual Desire as Part of a Dynamic Process

Sexual desire and behavior are central phenomena in PH that should be regarded as dynamic and everchanging processes. This is particularly the case for PH, where dynamic associations between emotion regulation and sexual desire or behavior can play an important role [2]. Therefore, we propose the Experience Sampling Method (ESM, [6]), also known as Ecological Momentary Assessment (EMA; [48]), to investigate hypersexuality. In ESM designs, there are usually 5 to 10 measurements per day for 5 to 7 days in a row, allowing for the construction of timeseries of the dynamic aspects under investigation. We propose that hypersexuality research makes use of ESM to gain better founded insight into PH. There are three advantages of ESM we would like to present: (1) ESM can improve construct validity of hypersexuality measurement; (2) ESM can improve ecological validity of hypersexuality measurement; and (3) ESM allows for the investigation of the associations between fluctuating sexual desire, sexual behavior, and other dynamic processes, be they emotional, physical, or social.

The first advantage of the use of ESM to assess hypersexuality is that it can improve construct validity of its measurement. If the validity of a test signifies that "variations in the attribute causally produce variations in measurement outcomes" ([49], p. 1061), one might wonder how the many variations in sexual desire that a person can experience throughout the day can be assessed with one-off cross-sectional methods. Sampling a sequence of experiences at the moment they occur is more like the real-time sampling throughout the day of, for instance, hormone levels. If these levels, or experiences, tend to change often, measuring them frequently might be optimal, and perhaps even necessary, to gain perspective on the patterns of change. Thus, if the construct of sexual desire is seen as something that is wont to fluctuate, the validity of its measurement will increase if these fluctuations are measured, too. This is in particular the case when investigating hypersexuality, where we can expect stronger associations between mood and sexual desire and behavior [50]. Therefore, we propose ESM as a method that might improve the construct validity of hypersexuality measurements because, with this method, the fluctuating nature of the phenomena under study is taken into account.

The second advantage of using ESM to assess hypersexuality is that it can improve ecological validity [7] of its measurement. Sexual desire and behavior are dynamic aspects of daily life and are part of an emotional and situational context. Contrary to experimental

settings, answering questions on a smartphone allows for real-time sampling of current experiences [51] without great intrusion on the participants' daily life. Especially in the case of hypersexuality research, experiments might take the participant a long way from emotional safety. When participants in an experiment afflicted by PH are asked to watch porn in a lab setting (e.g., [52]), they might experience shame they would not feel when watching porn at home, as they know the researcher will -one way or another–observe their actions. This could change the process under investigation drastically and disallow any conclusions on emotion dysregulation, sexual desire, and their reciprocal effects. Maintaining ecological validity by using ESM assures that the measurement itself will affect outcomes as little as possible [53]. Therefore, we propose ESM as a method that can improve the ecological validity of hypersexuality research, as it tracks participants' experiences while they go about their daily life.

The third advantage of ESM is that it allows for the investigation of fluctuating sexual desire, sexual behavior, and emotional, physical, and social dynamics. Within different previous conceptualizations of PH as sex addiction or hypersexual disorder, attention has been given to emotion dysregulation [2,19], or mood modulation [54]. The impact of mood on sexual desire has been studied outside of hypersexuality research as well [8,55]. As interactions between mood and desire typically play out over time, we suggest temporality should be included in research designs. This might for instance result in idiographic profiles of participants in ESM research. As an example, we show in Figure 4a the network representation of timeseries data collected with ESM of one participant. After analyses, the reciprocal impact of different states on one another can be pictured in a temporal (Figure 4b) and a contemporaneous (Figure 4c) network model [56]. The temporal network shows, among other effects, that for this participant, an increase in self-esteem predicts an increase in sexual desire some 90 min later. The contemporaneous network shows only a negative association between self-esteem and shame. Other analytical techniques can be applied, as well. With multilevel models applied to ESM data aggregation of results over individuals is possible and interindividual differences in intraindividual processes can be analyzed [57]. Differences between individuals might be predicted by group membership or other person level characteristics (e.g., [58]). ESM thus allows for the investigation of connections between emotion dysregulation and high sexual desire and behavior in PH. It can do this also in a relational context, if relevant (e.g., using actor–partner interdependence modelling, [59]). Emotion dysregulation and high sexual desire in PH have previously been presented as a vicious circle [60,61], in which out-of-control sexual behavior reinforces shame [62] or low self-esteem [63]. These negative mood states, in turn, reinforce PH behavior, thus giving the wheel another swing. These views have led us to incorporate emotion dysregulation and high sexual desire as drivers of PH. However, the theorized vicious circle has hardly ever been empirically investigated because cross-sectional and experimental designs are not well capable of assessing temporal aspects of PH while maintaining ecological validity. With ESM, it will be possible to test the vicious circle of emotion dysregulation and high sexual desire in PH, as it allows for the investigation of contemporaneous and temporal associations without disrupting the flow of daily life.

In conclusion of this third quark for hypersexuality research—assess hypersexual desire as part of a dynamic process —we reiterate that researchers should consider the changeable aspects of hypersexuality, in particular in the case of PH. Until now, sexual desire and behavior have methodologically been regarded as stable aspects, while they clearly are not. That these inherently changeable aspects have been treated as stable traits might be due to the limited possibilities of measurement methods rather than to the nature of sexual desire and behavior itself. Therefore, we propose to use ESM more in hypersexuality research, as ESM techniques can award attention to temporal aspects that might be key to understanding PH. We regard ESM as a complement to cross-sectional and experimental methods that can increase the validity of hypersexuality research and extend its scope to dynamic associations of emotion regulation, sexual desire and behavior in hypersexual populations.

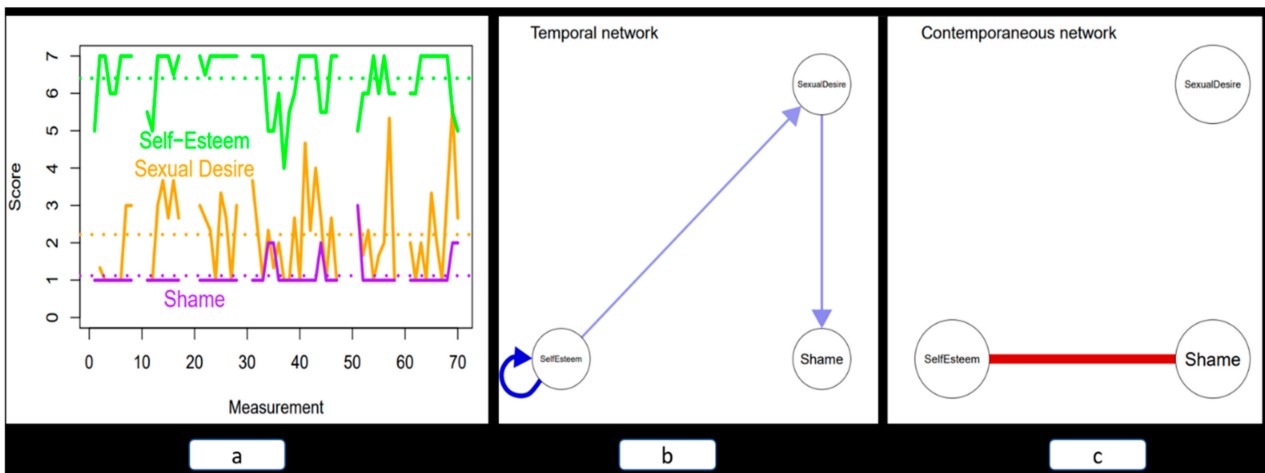

**Figure 4.** ESM output. (**a**) Representation of the timeseries of sexual desire, shame, and self-esteem of one ESM study participant. Data are collected over a period of 7 days, with a maximum of 10 measurements per day. (**b**,**c**) Nodes represent fluctuating feelings measured over time, and lines between nodes represent significant positive (=blue) or negative (=red) associations between these feelings. (**b**) The temporal network represents associations over time, with a time lag between measurements of 90 min on average. Note that self-esteem shows an autoregressive effect, which indicates that an increase in self-esteem 90 min previously predicts an increase in current self-esteem. (**c**) The contemporaneous network represents cross-sectional correlations. Note: Data are from participant 7 of the "CompulsivePornuseData.txt" dataset accessed at https://osf.io/jha8n/ on 8 December 2022. Participant 7 scored high on a measure of compulsive porn consumption, considered himself addicted to porn, and watched porn seven times a week.

We would like to present three considerations that might temper enthusiasm about the application of ESM. The first consideration pertains to the question: "Is recall really not good enough?" Bancroft et al. [55] importantly used evaluative recall in a survey addressing what usually happens to sexual interest and erectile responsiveness when depressed or anxious. Their method does capture dynamic aspects and is less burdensome for respondents than ESM. Others have noted that end-of-day recall [64], or even 7-day recall [65], of emotions or symptoms shows high correlations with real-time reports. Furthermore, we note that one-off measurement of sexual desire can provide important information, as it taps into the remembered self [51] and thus gives insight into how the person perceives and evaluates her/himself. This might not be far off the mark. However, for some processes, the individual will not be conscious of the patterns that are lived through. If these processes are the target of a study, ESM is still the best option. We think that this might be the case when investigating the vicious circle of emotion dysregulation in PH, where short-term temporal associations might distinguish PH from other conditions [58]. The second consideration: it is not clear beforehand what measurement frequency one should use. Often, an average time interval of 90 min is used, but this interval length is chosen rather arbitrarily. It is important to realize that different interval lengths implicate different constructs. Daily diary measurement—strictly speaking not ESM [51]—represents another perspective on sexual desire fluctuations than ESM, as the time interval used there is 24 h. However, a 90-min interval seems close to the ideal length when the target is the interaction between emotion regulation and sexual desire (e.g., [36]). The third consideration we put forth here is that there is a plethora of ways to analyze data collected with ESM. Especially in ESM research aimed at giving personalized feedback [66], differences in statistical analyses can lead to different outcomes and, thus, to different feedback [67]. As always, one needs to be cautious to specify too detailed models based on too little data. Furthermore, making research data freely available can address analytical heterogeneity, as this allows other researchers to apply their own analytical methods on the data. This might lead to a buildup

of ESM data on PH as well as to an exchange of analytical knowledge and experiences. To conclude this third and last quark for hypersexuality research, and despite the three caveats mentioned in this paragraph, we emphasize that making use of ESM techniques can improve the validity of hypersexuality research and provide valuable in-depth information on dynamic processes that underlie hypersexuality.

## 5. Conclusions

We have offered three quarks for hypersexuality research, and we hope that following up on our suggestions might serve to increase the objectivity of hypersexuality research. However, an increase in objectivity might come with an increase in complexity. If specific subpopulations need to be considered, one needs to determine and discuss what these are, keep track of different comparisons that are possible, investigate those, etcetera. The increased complexity when IRT or ESM are applied is evident, as these techniques require a higher technical level of data analysis than more basic designs. Nonetheless, we think this increase in complexity might be worth it, as it will come with an increase in nuance with regard to hypersexuality research. As there are still conflicting perspectives, nuanced views might benefit the field and settle long-lasting controversies. The heterogeneity in PH presentations, due to a manifold of sexual behaviors (e.g., pornuse or chemsex, [68]), might forestall essentialist perspectives on PH. Nuance might be the only option left [30,31].

Our three proposals for methodological improvement are linked to one another, as Figure 1 illustrates. Comparing relevant subpopulations can be important in both IRT and ESM. For the application of IRT in hypersexuality research, we suggest that comparing relevant subpopulations might even be essential if one seeks to determine what the most cue-valid characteristics are for PH. IRT is preferred over other methods to validate one-off measurement instruments to measure PH, and the use of contrasting samples to compare with is obvious here. When developing an instrument to discriminate PH from other conditions, one should include these other conditions in the validation process. Also in ESM research, the comparisons of different subpopulations can be essential in developing perspectives on PH (e.g., [58,69]). IRT and ESM are important complementary methods, as Figure 1 illustrates. One-off measurement instruments can focus on the stable and more permanent characteristics of PH as presented in our DCT description (https://psycore.one/construct/?ucid=problematicHypersexuality_7mm4hr4f, accessed on 31 January 2023). In the preliminary model for PH we show how these stable characteristics can come about when high sexual desire and emotion dysregulation drive the individual to repeat harmful, compulsive, and addictive hypersexual behaviors. The dynamic processes resulting in the more permanent outcomes might best be investigated with ESM, we have argued. The combination of IRT and ESM in hypersexuality research will provide information on permanent outcomes and on underlying dynamic processes that cause and maintain these outcomes. The application of relevant group comparisons in ESM and IRT studies will inspire, we hope, a deeper and more nuanced view on hypersexuality, both problematic and non-problematic.

We want to conclude our considerations with a reflection on negativity bias in sex research. This bias, we feel, has led to the neglect of positive sexual experiences, especially in the field of hypersexuality research. The possibility that–for some people–intense preoccupation with sex is a satisfying lifestyle has not often been considered (but: [20,70,71]). Early classifications of hypersexuality as the pathological condition of satyriasis or nymphomania [72] remained visible in the ICD-10 classification of "Excessive sexual drive", still in place less than a decade ago [29]. That excessive sexual drive has long been seen as problematic might have directed research away from an obvious question: is it? While on the one hand, this seems a matter of opinion [1,26,73,74], on the other hand, only empirical research (e.g., [14,20]) can hope to answer a question like this. Such research will be more independent of opinion if it is executed with a higher level of methodological rigor. Therefore, we hope that the changes in research methodology we propose will improve

the objectivity of research output and aid in reducing the influence of societal norms on sex research.

**Supplementary Materials:** The following supporting information can be downloaded at: https://www.mdpi.com/article/10.3390/sexes4010011/s1, File S1: IRT-tutorial.

**Author Contributions:** Conceptualization, P.V.T.; methodology, P.V.T.; software, P.V.T.; validation, Not Applicable; formal analysis, P.V.T.; investigation, P.V.T.; resources, Not Applicable; data curation, not applicable; writing—original draft preparation, P.V.T.; writing—review and editing, J.J.D.M.V.L. and P.V.; visualization, P.V.T.; supervision, J.J.D.M.V.L. and P.V.; project administration, not applicable; funding acquisition, not applicable. All authors have read and agreed to the published version of the manuscript.

**Funding:** This research received no external funding.

**Institutional Review Board Statement:** Not applicable.

**Informed Consent Statement:** Not applicable.

**Data Availability Statement:** Simulated data plus IRT tutorial available at osf.io/.

**Conflicts of Interest:** The authors declare no conflict of interest.

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
