# Peer review of "Three Quarks for Hypersexuality Research"

_sexes, doi:10.3390/sexes4010011_

Round 1

Reviewer 1 Report

Thank you for the opportunity to review an article titled ‘Three Quarks for Hypersexuality Research’. The Authors presented three important suggestions that may be applied to improve of research on hypersexuality/compulsive sexual behavior disorder: 1) considering relevant subpopulations, 2) using Item Response Theory to develop appropriate measurements, and 3) measuring sexual desire as it fluctuates in everyday life. I applaud the work put into these considerations. In my opinion, many of aspects described in this paper should be taken into account in future studies on CSBD. The manuscript is well written, the content is presented in an interesting and logical way. I have read this paper with a great interest! In my opinion, this article is suitable for publication in the Sexes.

Author Response

Dear reviewer,

Thank you for your kind words, we are very happy to receive your qualification of the article.

The authors

Reviewer 2 Report

1.     In section 2, paragraph 1, the authors should clarify the how and why regarding their statement about generating misleading results when instruments are constructed using a comparison of clinical PH to general population.

2.     Please clarify what makes a subpopulation “relevant” in the Introduction.

3.     Section 2, paragraph 2: It is not clear how the small percentage of individuals with PH prevents conclusions from being drawn about PH. The types of comparisons referenced by the authors are comparing sample characteristics to make statements about population differences and the generalizability of these findings would only be limited by degree to which the samples generalize to their larger subpopulations. As such, the limiting factor in drawing conclusions from these studies would be the n of each sample, not the difference between the two sample sizes.

4.     Section 2, paragraph 3: While the authors make an astute point, I do not find the point as relevant as claimed. For example, it is still true that high sexual desire is a useful screening tool for identifying PH as 100% of the participants with PH have co-occurring high sexual desire. So, while the relationship between sexual desire and PH is not significant in a sample with only high levels of sexual desire, that relationship is still useful for distinguishing those at risk of having PH from the rest of the general population because – based on visual inspection – it appears that ~90% of the general population does not fall into the high sexual desire group.

5.     The section about comparing relevant subgroups is vague and could benefit from a clearer thesis. For example, the authors could make a case that research should compare PH to NH or some other relevant subpopulation. Currently, the criticisms lack backbone because, without a clearly defined relevant subpopulation, comparing to the general population seems prudent.

6.     The authors’ concern about unnecessarily pathologizing individuals due to using instruments that have not been precisely calibrated is well taken and deserves to be more clearly stated and described.

7.     I appreciate the authors’ discussion of the benefits for using IRT to develop instruments and item banks and view this as one of the major strengths of this paper. Nevertheless, the discussion of this topic is unnecessarily lengthy.

8.     The authors’ suggestion of creating one dominant factor for evaluating PH directly conflicts with their own discussion of PH having two components: 1) pathological sexual desire and 2) emotional dysregulation.

9.     This paper’s biggest flaws is that it presumes a well-defined subpopulation of individuals with PH. We do not yet have an agreed upon definition of PH, and Hypersexual Disorder was rejected from DSM-5. Therefore, this paper seems to be ahead of the field in the sense that we cannot create well calibrated, valid instruments for measuring such a vague construct. Perhaps the authors could consider rewriting this paper to focus on Compulsive Sexual Behavior Disorder – a construct that, although not without criticism, has been clearly defined by the ICD-11.

10.  The paper simultaneously addresses too many issues and could benefit from increased focus and brevity. The ideas presented did not seem to fit together smoothly in many places. For example, the authors focus on methods for creating better measurements instruments at some points and what the defining characteristics of PH are at others. As noted earlier, we cannot create strong measurement tools if we have not clearly defined what we are measuring. Therefore, I encourage the authors to restructure this paper to first clearly define PH and second describe how best to develop tools to measure it based on its defined characteristics.

11.  There were a large number of typos and formatting inconsistencies in this paper. I gently encourage the authors to spend more time proofreading and editing before submission.

Author Response

Thank you.

The authors.
